**Mobility Control as State-Making in Civil War:**

**Forcing Exit, Selective Return and Strategic Laissez-Faire**

*Christiane Fröhlich\* and Lea Müller-Funk\*\*[1]*

*\*German Institute for Global and Area Studies (GIGA), Hamburg, Germany*
*\*\*Danube University Krems, Austria / German Institute for Global and Area Studies (GIGA), Hamburg, Germany*

**Abstract**

This paper addresses the question of how different actors govern mobility during civil war, and how mobility control and processes of state-making interact in such settings. Mobility in civil wars is often considered a political act by the various actors involved: Leaving the country can be perceived as an act of opposition, as can moving between territories which are controlled by different, opposing factions. Drawing on literature on strategic displacement and on migration politics and civil war as a form of state-making, as well as empirical insights from the ongoing wars in Libya and Syria, the paper identifies three mechanisms of mobility control in civil war settings: forcing exit, selective return, and strategic laissez-faire. The analysis reveals that all three mechanisms are employed by state actor(s), rebels, and militias alike, and depend on how these diverse actors perceive—and discursively construct—displaced populations: Forcing exit and selective return as a form of expulsion are directed towards displaced people who are perceived as threats and as undesired, while strategic laissez-faire is practiced towards displaced populations perceived to be unthreatening. We interpret the three mechanisms as ways in which actors in civil war settings attempt to manipulate a country's demography to their own favour in a process of state-making. We show that, in our case studies, forcing exit and selective return are common in spaces where there is a dominant governing actor. Where rebels intervene only minimally to maintain their monopoly over the use of violence, forcing exit and laissez-faire mechanisms prevail. The paper is based on fieldwork conducted between 2018 and 2021 in Syria, Lebanon, Libya, and Tunisia.

**10,856 without references and table**

---

[1] Both authors contributed to the paper in equal parts.

**Introduction**

How is mobility controlled when civil war challenges a state's power? What do mobility control practices during civil war tell us about state-making strategies of different political actors, and about how such actors are trying to consolidate power? These are the questions we address in this paper.

Civil wars present an opportunity to reconfigure pre-existing social contracts and power constellations. War often weakens, and sometimes destroys, state institutions; yet war also brings the emergence of local orders with different actors attempting to control mobility within a specific territory (Arjona 2014). Khalaf (2015, 37) suggests that during conflict, governance shifts from the state to other players at the local level, such as religious authorities, tribes, and clans. Such new actors can originate in earlier partnerships between the state and other social organisations that strive to exercise social control and provide protection and justice (Wang 2021). At the same time, conflict-affected nations "have histories of weak social contracts (or a once strong social contract that has degraded)" (Murshed 2009, 35). Actors in civil war settings commonly engage in strategies of state-making, either to re-instate the old order, or to create a new one.

Drawing on the literature on state formation, migration, and civil war, we show that civil wars open a window of opportunity for old and new actors to redefine who controls mobility, and how. Torpey (1997, 240) argues that states' monopolization of the right to authorize and regulate movement has been intrinsic to the very construction of modern states. Along the same lines, Zolberg (2008) has also interpreted migration policies as a key instrument of nation-building; he argued that the US not only set the conditions for citizenship, but also decided quite literally who would inhibit its lands.

We argue that during civil war, actors try to monopolize the "legitimate means of movement" (Torpey 1997, 240) and thereby claim the authority to determine who may move within and across their borders, as such movement is (seen as) inherently political. Leaving the country and moving between territories that are controlled by different, opposing factions is therefore perceived as an act of opposition by actors involved in civil war, irrespective of whether that was the intention. The same holds true for external and internal return to rebel- or regime-held regions. In fact, this even applies to immobility: staying in a rebel-controlled area can be seen as an act of resistance. This is one of the reasons why information on all forms of (im)mobility in civil war settings is highly sensitive, with reliable statistics often not available, resulting in inconsistent and patchy data. In Syria, for example, access restrictions and legislation targeting humanitarian agencies are hampering attempts to monitor IDP figures and movements (IDMC 2021).[2]

Mobility control also has a performative function, as it demonstrates the capacity to exercise force and take hold of a population which different actors lay claim to. Research has established that actors actively construct authority, including over mobility, and utilise it to achieve different political goals in civil war settings; Sosnowski (2020b) calls this "imposing authority" in this context (see also Hagmann and Péclard 2011; Kalyvas 2006). In this sense, and parallel to what Sosnowski (2020b) has written about ceasefires as tools of state stabilization/survival, mobility control can be understood as "a type of wartime order" (Sosnowski 2020b: 273) which can be used by a range of actors to shift and reshape a society in a way which includes some citizens while excluding others (Sosnowski 2020b: 274). This also means that processes of

---

[2] https://www.internal-displacement.org/countries/syria

state-making in civil war are hybrid, with local governance playing an important role (Stepputat 2018).

Importantly, state borders shift and sometimes lose their meaning during civil war, while newly emerging borders can gain importance or even substitute the old. Mobility control is thus not limited to the external borders of a state but extends to internal borders as well – a characteristic that civil war states share with authoritarian and totalitarian states, where such control is often to the detriment of particular, "negatively privileged" status groups (for example the Soviet Union, Nazi Germany or apartheid-era South Africa, Torpey 1997, 243).

Our understanding of how mobility is controlled during civil war and how this interacts with state-making, however, is still in its infancy. This analytical gap is linked on the one hand to the fact that findings from conflict studies, particularly regarding strategic displacement, have not been sufficiently brought into dialogue with the literature on migration politics and its role in state-making. On the other hand, it is related to insufficient data and the fact that internal and external displacement and return migration are commonly analysed separately. This is problematic since mobility in civil war settings is almost always multiple and includes internal and cross-border displacement as well as return trajectories across internal and external borders.

This paper combines literature on "strategic displacement" with research on migration politics as a form of state-making and as part of a wartime order. It draws on empirical material from two internationalized intrastate wars which have triggered mass displacement within and across borders—the civil wars in Syria and Libya since 2011—to build a better understanding of different practices of mobility control in civil war settings. Our analysis spans the mobility continuum from internal to external displacement to different return movements, and we focus specifically on practices, as this allows us to uncover localised and often informal measures which add to the formal, state-level, institutional spheres that are commonly at the heart of conflict and migration studies.

From our data, we derive three mechanisms of mobility control: forcing exit, selective return, and strategic laissez-faire. We interpret these mechanisms as ways in which different actors attempt to manipulate a country's demography to their own favour as a strategy of state-making, and we argue that they are deeply linked to how these actors perceive—and discursively construct—displaced populations. Forcing exit and selective return as a form of expulsion are practices directed towards displaced people who are perceived as threatening and undesired, while strategic laissez-faire is practiced towards displaced populations perceived as harmless. These practices vary widely, between the two cases but also on the sub-national level, and can overlap. Importantly, we do not claim that our typology covers all forms of mobility control during civil war across the globe; rather, it should be understood as a starting point for reflection to which future research can add.

In the following, we first conceptualise mobility control and state-making in civil war. We then present our research design and methods before we elaborate on the mobility control practices and the connected three mobility control mechanisms we have identified. This section focusses on mapping out the mechanisms rather than theorizing the causes of their variation and provides an empirically rich, micro-level, comparative account of how state and non-state actors in Syria and Libya have regulated the mobility of displaced populations. The fourth section analyses under which conditions which actors use which mechanisms by zooming in on two specific localities in Syria and Libya. The final part reflects critically on mobility control as state-making in civil war and identifies possible avenues for future research.

## 1. Conceptualizing mobility control and state-making in civil war

As outlined above, civil war states as well as insurgents have a strong interest in controlling the mobility of people residing in, entering, or leaving their territory. We define civil wars as intrastate armed conflicts which involve violence by and against a state, and at least one rebel group (Butler and Gates 2009; Böhmelt et al. 2019). As Staniland writes (2012, 246), civil war is a contest over the shaping of political order in a contested area, which brings them conceptually very close to state formation. Many analysts view civil war in fact as competitive state-building.

Migration studies have shown that people displaced by war are often simultaneously governed on and across multiple arenas which are characterised by overlaps and gaps, *ad hoc* responses, and enormous inconsistency—a pattern termed "implicit forms of governance" (Betts 2014). This includes policies relating to emigration, diaspora, immigration, or asylum, as well as policies regulating livelihoods and living conditions. Importantly, governing principles differ for internally displaced people (IDPs) and people displaced across state borders, as IDPs' rights derive from national law and/or international human rights conventions (Martin 2012). It is thus a complex interplay of state and non-state actors in multi-scalar—local, national, regional, and/or inter-/transnational—arenas in which mobility control is designed and practiced.

This 'mobility control complex' and how it is linked to state-making is still understudied, however, and especially in civil war states. On the one hand, regarding Syria and Libya, there is an abundance of studies on the politics of reception regarding Syrian refugees in Turkey, Lebanon, and Jordan (Dionigi 2017a, 2017b; Betts, Ali, and Memişoğlu 2017; İçduygu and Şimşek 2016; Janmyr 2016; 2018; Tsourapas 2019), and to a lesser degree, on Syria as a host state for forced migrants prior to the war (Chatty 2010; 2017; Hoffmann 2016). Some texts also focus on Libya as a host and transit state for migrants (Paoletti 2011; Phillips & Missbach 2017; Brachet 2016). Several studies show that in some host states for people displaced by civil war, not having any legislation might be a deliberate choice (Saghieh and Frangieh 2014)—termed by some scholars the policy of "no-policy" (Janmyr 2016) or of "strategic indifference" (Norman 2019). Other scholarship has focused on the political sociology of return migration, i.e., the role of states in shaping return migration through their policies or institutions (Hagan and Wassink 2020). Most of this literature, however, focuses on return policies of Northern receiving states, especially in "organized" or "forced return" contexts, such as so-called "assisted voluntary return" (Cleton and Schweitzer 2021, Alpes 2020) or deportations (Kalir & Wissink 2016). Some attention has also been given to return policies of neighbouring receiving states in the Global South (Fakhoury and Ozkul 2019; Şahin Mencütek 2019; Içduygu & Nimer 2020), and to the ways in which sending states control their citizens' political participation abroad and, in some cases, strip them of certain rights (Tsourapas 2018; Chaudhary & Moss 2019).

Migration scholars have also shown that controlling the movement of people across internal and external borders is crucial for state formation and consolidation (Torpey 1997; Torpey 2000; Zolberg 1978; Zolberg 2008; Vigneswaran and Quirk 2015, McKeown 2008). Yet what does this mean for a civil war context, which presents opportunities to redefine the boundaries of a nation and its territory? And how do different actors in civil war states control—and cooperate in—return mobility as one of the different facets of state-making?

Research from conflict studies has provided important insights into why people flee war (Adhikari 2013; Davenport, Moore, and Poe 2003; Moore and Shellman 2004), on refugees as political actors in civil war (Salehyan 2009; Bohnet et al. 2016), and on the logics behind combatants forcing people to flee, for instance to gain control of a territory, or to learn more about displaced populations (Lichtenheld 2020, Greenhill 2008; Hägerdal 2019; Steele 2018; Zhukov 2015). Conflict scholars have coined the term "strategic displacement", referring to targeted displacement to keep opponents from key resources (Zhukov 2015), or to displacement as a form of collective punishment (Balcells and Steele 2016; Steele 2018). According to Lichtenheld (2020), state actors have employed strategic displacement in two-thirds of civil wars between 1945 and 2008.

Steele (2019) identifies three types of displacement in civil war: individual escape (as a reaction to selective targeting), mass evasion (to avoid indiscriminate violence), and political cleansing (collective targeting based on shared traits like ethnicity, sect, political identity etc.). Lichtenheld (2020), with a focus on state actors, also identifies three types of strategic displacement in civil wars: cleansing, depopulation, and forced relocation. He argues that strategic displacement is not only a strategy by combatants to expel undesirable or disloyal populations, but also a strategy "to identify the undesirables or the disloyal in the first place", as they are forced "to send costly and visible signals of allegiance and affiliation based on whether, and to where, they flee" (Lichtenheld 2020, 3). Research on demographic engineering (Morland 2014; Teitelbaum 2015), finally, has focused on how ethnic groups deploy demographic strategies. While conflict studies have thus contributed important insights into the logics of exit, there is a gap in the literature regarding the logics of controlling return and the absence of mobility control during civil war.

Building on, but also departing from this, we study how mobility control and state-making interact in civil war settings. With regard to mobility control, we build on Zolberg (1978, 243) and Natter (2019, 31) to define it as (i) practices around formal policies, laws, and regulations governing internal and external border control, entry, and exit regulations; (ii) informal dynamics (for example, differences between administrations and localities); and (iii) laissez-faire and the purposive absence of regulation. The rationale for including the absence of regulation is motivated by our wish to understand implementation gaps, degrees of lawlessness, and legal vulnerability along the full continuum of movement in civil war settings. We also see the societal and political negotiation of displacement terminology—in other words, processes of labelling mobile populations—as part of mobility control, because different terms indicate different reactions to displacement (Zetter 1991; 2007; Brun 2003; Ottonelli & Torresi 2013; Erdal & Oeppen 2017).

In a second step, we need to establish the meaning of the state and state-making in (civil) war. According to Weber (1919), the state is a "human community that (successfully) claims the monopoly of the legitimate use of physical force within a given territory". States can also be understood as bureaucracies which collect taxes, produce statistics, and have military means at their disposal (Levi 2002; Tilly 1992). Clifford Geertz (1980) complemented this institutionalist perspective with a cultural approach on the more performative elements of legitimizing state power, such as rituals and norms. The question of 'Who belongs' is therefore crucial for consolidating the power of the state, especially during (civil) war: As Zolberg (2008, 11) writes, states have acted ruthlessly to push out religious, ethnic, or social groups they considered undesirable and incapable of being subjected or transformed. In fact, historically speaking, war, including civil war, is a more successful path to statehood than peaceful processes (Coggins 2011). Displaced people have therefore been 'produced' in processes of boundary drawing

through which nation-states are re-shaped or reconstructed since the early modern era. Violent conflict has in fact been theorized to be the main driver of state formation (Tilly 1978, 1992).

War often weakens, and sometimes destroys, state institutions; at the same time, Migdal (2001) has shown that the state is uniquely positioned to use compliance, participation and legitimacy to protect and consolidate its territory (Migdal 2001, 52-53). Yet, local institutions emerging in the midst of war can also produce order (Arjona 2014). In fact, armed groups have incentives to create new institutions as they may, for example, facilitate recruitment, provide access to political networks, allow for the accumulation of material resources, and even put their ideology in practice by implementing promised reforms. While the interplay of state, societal, and rebel forces as creators of institutions is certainly a characteristic of state-making in civil war (Arjona 2014, 1362), the dynamic process of negotiation and contestation of different actors to accumulate power and authority also needs to be taken into account (Sosnowski 2020a). Staniland (2012) argues that wartime political orders vary according to the distribution of territorial control and the level of cooperation between states and insurgents. He distinguishes between different civil war orders ranging from collusion and shared sovereignty to spheres of influence and tacit coexistence to clashing monopolies and guerrilla disorder.

From our point of view, mobility control in civil war contexts has two main functions: First, to set the boundaries of belonging, actors seek to define who is 'desirable' or 'undesirable', thereby reconfiguring the social contract between the state and its citizens. While this dynamic can sometimes work along ethnic, racial or religious line, it does not have to (Teitelbaum 2015). Second, mobility control also has a performative function, as it can demonstrate a challenged state's continued capacity to exercise force, but also a rebel group's capacity to behave as an independent state. For the latter, it has been shown that once control of a geographical location is established, other efforts to create a state follow, such as building up institutions, providing social services (McColl 1969; Stewart 2018) or humanitarian assistance to the displaced. Who provides for the displaced is often a key question of national sovereignty in conflicts (Rahal and White 2022).

Importantly, mobility control has deep roots and precedes civil war; logics of exclusion and of reconfiguring the social contract often exist long before a conflict breaks out, with our two cases being no exception. In Syria, violent repression, in particular against Kurds and Muslim Brotherhood supporters (Hama 1982), had been a decades-old practice. In Libya, political opponents of the former regime had long been persecuted in- and outside of the country (Nordheimer 1984; Dionne 1984). What is more, the Libyan state has a history of systematically discriminating against and excluding non-White and non-Arab minorities, such as Tebu and Amazigh communities. Qaddafi's Arabization policies focused on Libya being a white, Arab and Sunni Muslim nation, which was to be reinforced by erasing tribal and ethnic bonds. Such pre-existing dynamics are often accelerated by conflict, with previously established accountability mechanisms losing their meaning.

## 2. Research design and data

We apply a practice tracing approach to identify and characterise interactions between mobility control and state-making. Following Adler and Pouliot (2011), we understand practices as both contextually embedded and general patterns of action. Practices are both performative and "make other things happen" (Pouliot 2014: 241), giving them causal power that shapes society and politics: they produce very concrete effects in and on the world. Importantly, many

practices we have discovered in our data were not based on formal policy or public agreements, but were opaque, informal, and locally (re)negotiated. As one of our respondents put it for Syria: "I think the policy as such is not a universalizing policy, but rather a policy of deciding things not necessarily on an ad hoc basis but creating local solutions (…) [which] work in the government's favour" (SYREX4). This is why we use practice tracing, a hybrid methodological approach which establishes social causality locally by making sense of messy arrays of practices, but with an eye to producing analytical generalisations (mechanisms).

Methodologically speaking, the fact that practices describe "ways of doing things that are known to practitioners" (Pouliot 2014, 237) means that practices must be understood from within the community of practitioners to restore the intersubjective meanings that are bound up in them. In our case, the community of practitioners consists of policymakers and international stakeholders involved in migration governance and humanitarian aid, but our study also includes the perspective of the governed, i.e., displaced people, who have been subjected to this type of mobility control themselves.

To move beyond singular causality toward cross-case insights, we draw on mobility control practices from Syria and Libya as two examples of contemporary civil war states. Both can be understood as emblematic cases given the politically organized, large-scale, sustained violent conflicts that have occurred within their territories over the past ten years. They are characterised by an uneven state presence, with the Syrian regime continuing to govern after having re-conquered large, but not all parts of the country, and Libya consisting of two separate parts with competing governments after the overthrow of the Qaddafi regime. Both countries have experienced massive displacement within and across borders.

We identify practices and mechanisms in these two cases by drawing on three sets of data: First, a policy, document and press analysis including reports from NGOs and IOs; second, 88 narrative interviews with people who have experienced displacement (and to a lesser extent, people who have experienced immobility); and third, 31 expert interviews conducted with political actors, representatives of IOs, local and regional NGOs, and academics. Both types of interviews addressed how mobility within and across borders was controlled.

The data was collected between 2018 and 2021 together with a team of research assistants based in Syria, Lebanon, Libya, and Tunisia. Large parts of the fieldwork took place during the Covid-19 pandemic, so that many expert interviews had to be conducted online. Also, some narrative interviews in Syria and Libya had to be conducted virtually. All interviews have been anonymised and are quoted with pseudonyms and codes (see annex for more information about our sample).[3]

### 3. Mechanisms of Mobility Control in Civil War Settings: Forcing Exit, Selective Return and Strategic Laissez-Faire

In the following, we present the results of our practice tracing exercise across our two cases. The mobility control practices we detected were not only applied by state actors; rebel groups and other actors challenging the state also engaged in them. We have grouped the mobility control practices into three mobility control mechanisms: forcing exit, selective return, and

---

[3] Narrative interviews: SYR = interviews with Syrians inside Syria who have experienced immobility, displacement and/or return; LEB = interviews with Syrians in Lebanon who left Syria in the context of the war; LIB = interviews with Libyans inside Libya who have experienced immobility, displacement and/or return; TUN = interviews with Libyans in Tunisia who left Libya in the context of the conflicts; expert interviews: SYREX = expert interviews about Syria; LIBEX = expert interviews about Libya.

strategic laissez-faire, in a process of theoretical abstraction (see Table 1). Within each mechanism, we have grouped the different practices into families for analytical purposes.

In both contexts, these three mechanisms were accompanied by the discursive construction of 'unwanted elements'. In Syria, branding displaced people or entire regions as 'terrorists' or 'defectors' has become a common practice since 2011. As one of our respondents explained:

> "the regime sees us or sees the refugees and IDPs and most of the Syrian people actually as an enemy, actually in the last speech of Bashar Assad, he mentioned that 'It's very good that we get rid of the enemy'. (…) [To] many of the people there, I mean, within the regime, we are viruses, we are traitors, we are germs, etcetera, cancer. And also the regime, from the beginning, they have had a vision. They think—and * they announced it, they didn't hide it—they say that they want a 'useful Syria'. (…) General 'Issam Zahr el-Din for example, on the state television 2017, he warned the refugees never to set foot in Syria again; and he said exactly "We will not forgive them, and never forget what they have done". And also the head of the Air Force intelligence, Jamil Hassan, in 2018, he said that "the regime only wants loyalists", (…) "A Syria with ten million trustworthy people obedient to the leadership is better than a Syria with thirty million", so you know that they are thinking of refugees in that way." (SYREX11)

And indeed, in one of his public speeches, Bashar al-Assad said that Syria had gained "a healthier and more homogenous society" through the exile of Syrians (RT, July 2017). He also repeatedly referred to IDP locations as hotbeds for "terrorists", and to internal displacement as a result of "terrorism" (IDMC 2014, 13). Another respondent also implied that refugees outside of Syria were seen as a threat, as not (sufficiently) subordinate, resulting in them facing "worse institutions" even after the major war activities had ceased (LEBEX16).

In Libya, similar practices can be observed but in a less encompassing way: On the one hand, anti-Qaddafi militias branded people from entire tribes or regions as "henchmen" of the old regime (*azlām*), which became a term of accusation and slander in Libyan society and media (Jaidi and Tashani, 2015; TUN22, TUN25). Displacement across conflict lines in Libya is often interpreted as political affiliation. One Libyan respondent said that "people from the East displaced to the West or to Turkey are perceived as enemies in the East" (LIB1), others mentioned "hate speech against those who have left" (LIB3). On the other hand, Libyan political actors have also used depoliticised terminology to refer to Libyans who have experienced displacement. For example, internally displaced people were usually labelled as *nāziḥ*; the verb *nazaḥa* in Arabic refers to the act of going away from home, moving to other lands, but also to being displaced (in war), without a specification of whether displacement happens within or across borders, or of a legal status. A representative of the Libyan Ministry of Displacement and IDP Affairs also called externally displaced Libyans "Libyan migrants" (LIBEX4), thereby depoliticising displacement and downplaying protection needs, a tendency observed also in official documents addressing the situations of Libyans who fled the country.

*Table 1: Mechanisms and practices of mobility control in civil war states*

| Mechanism 1: Forcing exit | | Type of actors |
|---|---|---|
| *Practices of extinction* | Torture, killing, bombings, besieging | State actors and insurgents, national and local, weak and strong |
| *Practices of deprivation* | Hindering access to aid, healthcare, education, food, water through border closures/control | State actors and insurgents, national and local, weak and strong |
| *Practices of irregularisation* | Hindering access to legal documentation, HLP rights, irregularising most cross-border movements | Mostly state actors, national and local, strong |
| *Practices of cross-line deportations* | Deportations across conflict lines | Mostly state actors, national, strong |
| **Mechanism 2: Selective return** | | |
| *Practices of arbitrariness and informality* | Frequent and intransparent policy changes, unreliable implementation, disinformation, corruption and nepotism | State actors, national and local, weak and strong |
| *Practices of deprivation* | Hindering access to private property, rights-stripping, fining | State actors and insurgents, national and local, weak and strong |
| *Practices of "taming"* | Security clearances, reconciliation agreements | State actors, national and local, strong |
| *Practices of threatening potential extinction upon return* | Sending to the front, detainment or killing upon return | State actors and insurgents, national and local, weak and strong |
| **Mechanism 3: Strategic laissez-faire** | | |
| *Practices of non-restriction* | Free movement for people considered loyal to governing actor | State actors and insurgents, national and local, weak and strong |
| *Practices of non-protection* | No HLP rights, no help in case of retaliation, no material support in case of return | State actors and insurgents, national and local, weak and strong |

### 3.1. The mechanism of forcing exit

The objective of this mechanism is to force certain groups which are perceived to be unwanted elements of a future state to leave territory through strategies of immobilisation and existential threat. One family of practices in this category are those primarily aimed at extinction. These include the most extreme and violent practices mentioned in our data, such as torture, targeted killings, bombing and besiegement: In Syria, government sieges and bombings of rebel strongholds have been a significant driver of immobilisation but also in-line (internally, within conflict lines), cross-line (internally, across conflict lines), and cross-border displacement (across international borders)  throughout the war; in some areas, they produced high death rates among trapped populations, while in other areas, they resulted in a population exodus. Also, the practice of targeting civilians in opposition-controlled areas forces exit by making such areas unliveable (IDMC 2014, 10). Imprisonment, forced disappearances and torture can equally be seen as practices of forcing exit by infusing existential fear in those who survive and witness it. As one Syrian respondent said: "the mass arrests, security crackdowns, the harassment, forced disappearances, systematic siege, starving, destruction. And now

confiscation of lands and demolition of buildings, naturalization of foreign militias. (…) from these points you know that's how the regime is thinking about refugees and IDPs" (SYREX11). In Libya, different militias have been terrorising civilians over their alleged political affiliation through beatings, shootings, and brutal treatment during detention.

Another family of practices revolves around depriving populations of aid, healthcare, education, food, water and other services and goods. This includes practices of border control, where different actors restrict access to areas across both front lines and state borders, resulting in international humanitarian agencies being unable to provide aid and services, and in supply difficulties, which in turn force those to leave who cannot survive without such aid. In Idlib, for instance, borders with Turkey have been closed since 2015, which our respondents described as the area turning into a prison, the population being cut off, and no one coming in (SYR6, SYR7, SYR11). Similar dynamics can be discerned for restrictions on cross-line movement, too, which has become almost impossible for humanitarian actors wanting to provide aid to IDPs inside Syria: "Currently, the crossings are closed between the areas of the Syrian regime and the areas of northern Syria, liberated from the regime's control" (SYR15). A respondent from Damascus described the borders between government-controlled (goS) and rebel-held areas in Syria in 2020 as follows: "I have the feeling that parts of the country are cut off. (…) The difficulties were when you were cut off from the rest of the country. That is, you are locked up from outside" (SYR20).

A third family of practices aims at irregularizing and criminalizing those moving within or across frontlines and borders. This family includes practices by state actors and rebel groups of controlling internal movement through checkpoints, but also of preventing IDPs from acquiring legal documentation, like ID cards or birth certificates. This has been the case for many IDPs in rebel-held areas in Syria or those with a "security sign" attached to their name from, (SYR19, SYR6, Danish Refugee Council & Danish Ministry of Immigration and Integration 2019: 13-16). Furthermore, fleeing civilians can also be subjected to a process of registering and screening before authorities allow them to enter IDP centres, for instance in Eastern Ghouta; many IDP shelters have in fact operated as detention centres (EASO 2020, 22). In Libya, respondents also talked extensively about dangers at checkpoints and their fears of being recognised or being persecuted by militias because of their family name or their tribal affiliation. One interviewee referred to the main road from the south of Libya to Tripoli, for example, as "suicide road" for this reason (TUN6).

Irregularization and criminalisation also happens at external borders, where practices of border closures and control allow certain groups to pass, but not others, thereby irregularising the latter's movements. In Syria, the increasing border closures with Syria's neighbouring countries, in particular Turkey, Lebanon and Jordan, criminalised large parts of those fleeing the country from 2015 onwards. At the same time, INGO staff, for instance, can request special permission to cross, as can those working for key local councils or the Syrian interim government. Movement across the border for medical reasons is also still allowed, and Turkey relaxes the border closure on high holidays to allow families to reunite. All other movement is considered irregular and a criminal offense. This practice depends in large parts, from the point of view of Syrian actors, on cooperation with state and non-state actors in neighbouring countries, often based on their involvement in the conflict. This is most effective when political interests align; for example, in Lebanon, the change from open mutual movement between Syria and Lebanon to the closure of the border was supported by the Lebanese government, including Hezbollah, al-Assad's powerful ally.

Fourth, practices of cross-line deportations aim at forcing exit across conflict lines. Examples from Syria include people from rebel-held areas who were deported as a result of so-called "reconciliation agreements" between regime and insurgent forces after or shortly before the regime reconquered territory. The regime's negotiators offered different kinds of deals in different areas; for example, those that demonstrated high resistance in fighting the regime faced total population removal and safe passage to rebel-controlled areas (Hinnebusch & Imady 2017, 7), such as deportations from Rif Damascus to Idlib (SYR5), or from the last part of opposition-controlled Eastern Aleppo to opposition-controlled Idlib in 2016. The famous green buses which shuttled people have been used for all deportations and have become a performative symbol of opposition defeat (Barnard and Saad 2016).

### 3.2. The mechanism of selective return

The mechanism of selective return aims at sorting and controlling returnees as a sort of selective expulsion by repelling some groups of displaced people from returning, by re-subjecting others under the control of state and non-state actors. Within this mechanism, we discern four families of practices: Practices of arbitrariness and informality, practices of deprivation, practices of "taming", and practices of threatening potential extinction upon return.

The first family of practices is situated at the gap between official policy offering safe return and practices on the ground, which concerns mostly state-actors. For example, the Syrian regime has been trying to foster an image of stability since reconquering large parts of the state territory, among others by repeatedly calling on refugees to return. A number of decrees and regulations have been passed in this regard, including the waving of some fees and fines for late registration of vital live events or for border crossings, as well as decrees around conscription (SYREX2). Also, external actors have supported the return narrative; Russia in particular has been helping in this regard, for instance with a jointly convened Refugee Return Conference in Damascus in December 2020. At the same time, the Syrian government as well as other political actors have been enacting informal practices to repel returnees, signifying a lack of a clear strategy on return, and a high degree of arbitrariness, where decisions tend to depend on individual officers. This applies to both internal and external return movements. As one respondent put it: "All governing actors have a return policy: goS, SDF, HTS, but we see no return, so (…) they are completely failing at implementing their return policy, which makes you question – do they really have a return policy, or have they just written a policy to appeal to their patron, or to their sponsor? So, does the Syrian government really want Syrians to return, or are they just doing it because Russia told them to do it? And does the Syrian opposition really want Syrians to return to its areas or are they doing it just because Turkey told them to do it?" (SYREX9). Practices in this family include frequent and untransparent policy changes regarding return, unreliability of how returnees will be treated at internal and external borders and checkpoints (Danish Immigration Service 2019), practices of disinformation, for instance contradictory reports on who needs to apply for a 'reconciliation agreement' (EASO 2020b), as well as corruption and nepotism.

Practices of deprivation include hindering access to private property, rights-stripping, and fining upon return, which is another strategy undertaken by state and non-state actors to repel potential unwanted returnees. An example for this family of practices is the arbitrary demolition of private property in former rebel-held areas for 'state reconstruction projects' (SYR10, SYR19, SYR20; EASO 2020, 34), as it *de facto* regulates return to those areas. They also include rights stripping, especially House, Land and Property (HLP) rights, with the Syrian Law 10 being a prominent example (Abu Ahmad 2018). This law essentially legalised the

expropriation of those Syrians who cannot or are not allowed to return to their property in government-held areas, illustrating how different practices can interlink to produce a certain outcome, in this case practices of irregularisation and of deprivation. In Syria, it has also become common practice that returnees must exchange 100 USD into Syrian pounds at the border at a conversion rate set by the government, thus essentially "preying upon those vulnerable families who are often returning because, you know, they've exhausted their resources and they have no other options" (SYREX4). This repelling practice also extends to internal return: Internal returnees are also prevented from accessing their homes and property by government checkpoints in some (usually previously rebel-held) areas (SYREX5). One example are checkpoints in formerly opposition-held areas in South Damascus, the purpose of which is to control the flow of individuals entering and exiting Damascus, particularly people considered 'undesirable', i.e., former residents of Eastern Ghouta and other opposition-held districts outside of Damascus (Danish Immigration Service 2019, 13-16).

Practices of "taming", in contrast, refer to security clearances and "reconciliation agreements" required for return across conflict lines and international borders. Syrian refugees who want to return to Syria from outside the country, for example, need to apply for a security clearance or "sorting out of affairs" (*taswiyat al-waḍaʿ*; EASO 2020b, 18) issued by Syrian security services prior to relocating and in some cases, a so-called reconciliation agreement. First, the authorities check whether a person has a "security issue". This goes back to a decades-old practice of the ruling Syrian Baath party, which has long relied on a system where citizens report on each other to security agencies. As a result, approximately 15% of all citizens reportedly have security issues. Second, refugees need to go through a so-called "reconciliation" process if their name is on a blacklist. Reconciliation committees consist of officers representing the regime's security branches, dignitaries, clerics, and officials from the region, which are also partially appointed by the regime's security branches. During this process, potential returnees have to share extensive personal information with the security apparatus, which then reportedly uses such data to blackmail or arrest individuals who are perceived as a "security threat" (Alpes 2021, 18-20). According to some sources, several Syrian refugees in Lebanon have already been denied security approval and could therefore not return. A report on this topic states: "An international organisation in Syria assumed that those who were not allowed to return were those whose names were on wanted lists. They were not allowed to return to Syria as the government wanted to send a message that those affiliated with armed opposition groups are not welcome in Syria" (Danish Immigration Service 2019, 24).

Practices of taming necessitate strong institutions, which makes it a practice mostly employed by strong(er) state-actors. Practices of taming in the context of cross-border return also require the cooperation with the host country – which is only possible if interests align. In Lebanon, non-state actors, particularly Hezbollah, have negotiated the "repatriation" of Syrian refugees with militant groups, without direct involvement of the Lebanese government or the United Nations. These steps were, however, in line with the view of the Lebanese government on "repatriation": "The Lebanese want Syrians to go back asap; the government calls this its refugee policy. They say: 'Repatriation shouldn't be conditioned, we don't want to wait for a peace agreement for people to return; having said that, return should be informed and voluntary'." (LEBEX12). Hezbollah, in cooperation with the General Security Office (GSO), also opened several offices across the country for refugees to register for return and to perform security clearances.

A fourth practice employed by state and non-state actors in Syria and Libya is threat through potential extinction upon return. This includes forcible conscription and subsequent sending to the front or detention upon return, but also threats to be killed upon return. In Syria, there is

strong evidence that people returning to government-held areas were conscripted and deported to the frontlines of the conflict by the regime; as one respondent explained: "Returnees are very very disinclined to indicate that they are refugee returnees […] but again, it is related usually to forced conscription, fears of forced conscription or detention in government-held areas. I know anecdotally—but I mean I've heard this from [name of organisation] for instance, so like anecdotes but not really anecdotes—that, like, IDPs who have returned to government areas as well as to a lesser extent refugee returnees who end up getting conscripted, are sent to Idlib and they're put right on the frontlines; it's absolutely retribution and punishment. […] That is relevant for refugees but particularly IDPs" (SYREX4). Also, some of the Libyan refugees interviewed in Tunisia talked about their fears of being recognised and killed by different militias upon return.

### 3.3 The mechanism of strategic laissez-faire

Finally, we have discerned the strategic absence of legislation as another mechanism of mobility control in civil war settings. This mechanism refers to practices which aim at allowing the movement of some groups, and at preventing that of unwanted groups by providing insufficient protection. The term *strategic* laissez-faire highlights the role of intentionality, which we define as "being aware of one's action causing harm, and (not) acting nonetheless" (Tyner and Rice 2016, 48). Laissez-faire thus should not be seen as a mark of indifference, but as one tool of many in the complex array of disparate regulations and practices of mobility control. Strategic laissez-faire is the least costly mobility control mechanism and necessitates little institutionalisation or capacity, which makes it a mechanism easy to employ by weak and strong state and non-state actors alike. Efforts mainly revolve around masking the selective character and the intentionality of these practices. For Libya, one respondent outlined how both governments do not deal with IDPs in practice, leaving this task to NGOs or private families and tribal structures instead (LIBEX5). Yet, it can also be a welcome choice of action for strong state actors. In Syria, a respondent characterized the government's partial reluctance to fulfil key state functions of mobility control and protection as "unfortunate non-enforcement", which can take the form of "bureaucratic impediments", but also of bribes, difficulties to get official documentation, or a lack of transparency on due process with regard to mobility control: "there is absolutely no specific mechanism that has been created to be able to monitor and use it as a recourse" (SYREX7). We have identified two families of practices in this mechanism: practices of non-restriction and practices of non-protection.

In both our cases, governments and other political actors have allowed in-line, cross-line and cross-border (exit and return) movements for certain parts of a population, in what we call practices of non-restriction. For instance, Syrians who are perceived as either neutral or pro-government do not require to sign the afore-mentioned reconciliation agreements, and some also returned without a security clearance (SYR12, SYR13, SYR15, SYR16, SYR18). In Syria's rebel-controlled Northwest, respondents described that they could move rather freely within rebel territory. In Libya, in contrast to militias, state actors left the exit of its citizens rather unobstructed: With some short exceptions (Covid-19, prolonged border closure of Ras al-Jedir in 2015), the Libyan-Tunisian border has, despite its increasing securitisation, remained open to Libyans. Libyan interviewees – with few exceptions – reported to have been able to cross the borders to Tunisia and Egypt rather easily and not having to resort to irregular means. Having said that, border posts are manned by different militias, which impacts displacement patterns. For example, in the aftermath of the toppling of Qaddafi, Qaddafi supporters mainly used the Ras Jedir crossing controlled by the Amazigh city of Zuwara, while opposition supporters crossed through Dehiba-Wazen, which was controlled by a local militia (Kartas 2013; UNHCR 2013a). In East Libya, opposition-affiliated militias controlled the Libyan-

Egyptian border post in 2011 (TUN1). Thus, the choice of border crossing offers some insight into perceived or real political affiliations of those crossing the border.

The other side of such practices of non-restriction are what we call practices of non-protection. For instance, the Libyan state(s) have not at all systematically addressed the issue of return, including those of former Qaddafi supporters. There have been hesitant attempts to develop a policy; for instance, a group of parliament members visited neighbouring countries to talk to Libyans and convince them to return, and in Cairo, the Libyan Embassy opened a separate office to listen to the problems of displaced Libyans. But nonetheless, a representative of the Libyan Ministry of Displacement and IDP Affairs highlighted that the ministry takes limited responsibility for Libyans abroad who have political views that are not in line with the current government: "we are in dialogue with Libyans abroad, some of them have political demands and ask for safe corridors. Unfortunately, the ministry cannot help with security corridors for all migrants, especially those with a political view, those who participate in demonstrations, who want to participate in political demonstrations, those supporting the previous government" (LIBEX4).

In fact, many of our Libyan respondents who were perceived to be affiliated to the former regime either felt misrepresented by or afraid to contact the embassy. A respondent whose family name suggested an affiliation to a pro-Qaddafi tribe explained: "When I returned to Libya, they found me, tied me up and beat me, they tortured me and broke my hand. My hand is filled with iron sticks now and I have all the papers to prove it, I underwent surgery twice and up until last week, I kept having issues with those who beat me up" (TUN22). Many also highlighted that there is no state support for damaged or looted property which could support their return; instead, they reported the uncontrolled looting and appropriation of their properties, sometimes with the explicit knowledge of state actors: "There are properties which were taken in 2014, we had lands in [street name] and [place name]. They were taken by people we call 'Guardians of the Blood' [a militia], and people who are considered legalized militants who belonged to the parliament in Tubruq, and we tried to get them back by law, but of course weapons spoke louder than the law till this very moment. They still rob people of their properties; they hunt for properties" (TUN2). In general, our Libyan respondents also reported a complete lack of support after private property was destroyed through war action: "[the] state promised reparation but [it was] all lies" (TUN10), "when [I] returned, the house needed to be fixed, but there was no support, there are still no services, I received only family support, no NGOs or IOs" (TUN12); "my house was looted and destroyed, I repaired everything, the government didn't help, there was no electricity, no water, no services upon return" (TUN13). In this policy vacuum, displaced people rely on family and tribal support, local communities and charities, and – to some extent – the support of international NGOs.

In the following, we illustrate under which conditions which actors use which mechanisms and practices of mobility control and how they interact with strategies of state-making in two exemplary vignettes, Jabal al-Zawiya in Syria, and Tawergha in Libya.

## 4. Explaining differences in mobility control

Zooming into the varied practices employed by a range of actors in our two localities, we expect variance both between the two cases as well as on the sub-national level in terms of civil war order and mobility control. In both cases, a diverse set of actors, including state, rebel forces and militias, have attempted to govern to a different extent and with varying success, depending on, among others, the course of the two respective civil wars. In consequence, mobility control

practices and related institutions, which were created by an interplay of state, insurgent and societal actors, have changed over time.

In Syria, where the regime survived, strategic laissez-faire is less likely, since identifying and expelling insurgents is a key mechanism of regime survival in such a situation. Different actors in Syria have thus mostly employed practices of forcing exit and selective return. The Syrian regime has only employed strategic laissez-faire for specific groups of displaced Syrians – those perceived as unthreatening. In Syria "rebelocracies" such as Idlib Governorate or Rojava exist – situations where an armed group becomes the *de facto* ruler in the broad sense; yet other sources of authority including state institutions remain present to some degree (Arjona 2014). In Syria's rebelocracy of Idlib, we can observe a civil war order of what Staniland (2012) called "clashing monopolies" (Staniland 2012). This order is characterised by violent competition between state actors and insurgents who control distinct territory. In Northwest Syria (NWS), limited military capability and agreements brokered by external forces, in this case Turkey and Russia, have led to a form of shared sovereignty, with Hay'at Tahrir al-Sham (HTS, Organisation for the Liberation of the Levant) having imposed itself as the dominant actor, but at the same time delegating governance tasks to other actors (al-Kanj 2021), including international organisations and NGOs (SYR11, SYR6).

In Libya, strategic laissez-faire practices can be expected to be more frequent, as well as some practices of forcing exit. We expect less frequent selective return practices. This is related to how the regime change played out in the country, with the overthrow of the former regime and two competing state actors now governing two separate parts of the former Libyan state. Because both governments have their own territory, it is less important to identify the opposition; rather, anti-Qaddafi actors put emphasis on identifying supporters of the toppled Qaddafi regime out of fear of a counter-revolution. At the time of writing, Libya's war time order could be characterised as tacit coexistence (Staniland 2012): the two Libyan governments cooperated passively in the sense of an acknowledgement that neither side has either the power nor the will to attack and dominate the other, necessitating a mediated mutual survival. In Libya's East and West, civil war orders could be described as "aliocracy" (Arjona 2014, 1375): Rebels only intervene minimally, not more than what is needed to maintain their monopoly over the use of violence. At the same time, other matters of governance and control have been delegated to other local actors – be it state officials, traditional authorities, civic leaders, or other.

### 4.1 Demographic engineering in Jabal al-Zawiya

Jabal al-Zawiya is an enlightening example of mobility control in a rebelocracy in Syria. Jabal al-Zawiya is a mountainous region in Idlib Governorate in NWS, with the biggest towns being Rīḥā and Maʿarrat an-Nuʿmān. It has remained one of the strongholds of the Syrian opposition since 2011. It was shelled and raided by the Syrian Army in 2012, experienced a siege in 2015 when "the area turned into a prison" (SYR6), and was still being bombed at the time of our fieldwork (2020). At the time of writing, the area was controlled by a rebel government under Islamist Hay'at Tahrir al-Sham (HTS) (Yūsuf 2021). When HTS won the inter-factional war with several other armed groups in North-West Syria, it changed the landscape in terms of who had both security and administrative control over the area (SYREX2).

Jabal al-Zawiya is a textbook example for practices of immobilization, forcing exit and selective return by state-actors and for practices of immobilization by rebel groups: The Syrian regime has persecuted and killed activists in the region, bombings have destroyed 90 per cent of the

urban and civil infrastructure such as hospitals, schools and universities, the region was subjected to a government siege, and internal mobility control is exercised through checkpoints, the blocking of humanitarian assistance, and massive depopulation campaigns as well as cross-line deportations. Respondents reported having been displaced within the region of Jabal al-Zawiya itself, to other areas within Idlib Governorate, but also to Afrin in neighbouring Aleppo Governorate and across the borders to Turkey and Lebanon, with some in-line and cross-border return movement to Jabal al-Zawiya. As one respondent from the region put it, "After Ramadan, we returned to the area of Jabal al-Zawiya, the population was 200-220,000 people before the displacement and after I returned, almost the entire population of the area was not exceeding 100 (…). Now about ten percent of the population have returned to the area" (SYR11). At the same time, the region has become a refuge for IDPs from former rebel-held areas across Syria and has experienced a lot of in-line displacement.

With the closure of the borders to Lebanon and Turkey, Jabal al-Zawiya's population has become increasingly immobilized. In 2020, the region was surrounded by regime forces on three sides, with one remaining northern corridor for evacuations (SYR11). Rebel forces also restrict the exit movement of civilians from Idlib Governorate. As one interviewee explained, both Hay'at Tahrir al-Sham and to a lesser extent the Turkish armed forces were very restrictive at checkpoints around Idlib: "if you want to leave as an IDP from HTS areas to self-administration-controlled areas, they levy quite large fees that are completely informal, they'll just take what they want from your car" (SYREX4). Few can afford to pay a smuggler to leave for Turkey, which in 2020 required almost $ 1,000 per family – an absolute fortune. For others, staying against all odds equals resisting a regime they loathe.

The case of our respondent Karim (SYR6) illustrates how the mobility of perceived 'enemies of the state' is being restricted by the Syrian regime in line with the above-mentioned strategies of state-making. He was born in Jabal al-Zawiya in the mid-1980s and was working in a ministry when the demonstrations began in 2011. His family was known to support the opposition, so he had a 'security issue' attached to his name since the very beginning of the conflict. This made it extremely difficult for him to pass the military checkpoints which abounded in the area. It was almost impossible, for instance, to drive his sister to hospital when she was about to give birth. In 2012, during the government shelling, Karim fled Jabal al-Zawiya to Turkey to join other activists and to work with an international NGO providing aid to Syrian civilians. While the Syrian-Turkish border itself was still open and easy to pass, it took him almost two days to circumvent Syrian Army checkpoints on the way to the border. But he returned to Syria after just one year, and by crossing directly into rebel-held territory, as he wanted to support the population in his region, safeguard his property, and ultimately considered staying an act of resistance: "If we left, we would have to leave our lands and fortunes here and we would have been labeled as displaced." Also, properties in Idlib had lost in value: "The same is true of real estate that is marked with signs that they are owned by 'terrorists'. Even if we can take it [the property] back, we lose a lot. (…) Here in Idlib, the lands are all falling in price because we are considered terrorists." Since then, our respondent has stayed in Jabal al-Zawiya working for NGOs assisting civilians and IDPs. He has been unable to obtain official documents for his son due to the 'security issue' attached to his name: "Since 2011, there are mainly people with no registration, there were people who went to regime areas to register their newborns but many were very afraid to do so." In this way, Karim and his family are slowly excluded from the Syrian society and state.

Jabal al-Zawiya was portrayed by our respondents as a region run by parallel institutions, caught between regime, rebel and Turkish forces, competing over who controls mobility and who decides who belongs. "We pay for different items in different currencies. We pay for gas and fuel in Turkish, which in turn is fluctuating against the USD. (…) As for pricing, in our region

there is a 'rescue government' and an interim government in the Afrin regions, they are setting the prices. But there are also long arms of the regime everywhere" (SYR6). On the other hand, the region was completely deprived of public services: "Education is not available here because the schools are destroyed. Now there are no public services of any kind, the nearest hospital is 38 km away from us in Idlib province. (…) There are no human rights organizations or international organizations that are providing assistance because we live in an area of direct conflict" (SYR11). According to a report from September 2019 published by Refugees International on the situation in Idlib, two thirds of the population of Idlib need humanitarian assistance (EASO 2020a, 15).

Some of our interviewees and other observers described the mobility control practices in Idlib Governorate as "demographic engineering" (SYREX9; SYACD 2020), a state-making strategy which relies on emptying government-controlled areas of perceived political threats by displacing them to rebel-controlled areas, and by depriving inhabitants of rebel-held areas of services, rights, and sufficient means to survive: "It's (…) an intentional policy of demographic engineering, for lack of a better word, and Bashar al-Assad alluded to it in one of his speeches where he talks about the homogeneity of society, and that he wants to create a homogeneous society. So it's definitely an engineering, a demographic engineering exercise that the Syrian government is involved in, trying to redesign the demographics of the areas under its control, and they will continue to do that as they are recapturing areas" (SYREX9).

### 4.2 Extinction, cross-line deportations and non-protection in Tawergha

The town of Tawergha, located in northwestern Libya, south of Misrata and 240km east of Tripoli, the capital of West Libya, could be categorised as as an aliocracy. Its population has been exposed to practices of extinction and cross-line deportations by non-state actors and practices of non-protection by state actors. The majority of Tawergha's population used to be non-Arab Libyans, many of them descendants of African slaves brought to Libya in the 18[th] and 19[th] centuries. During the Libyan uprising of 2011, pro-Qaddafi forces used Tawergha as a base for attacks on Misrata when they besieged the city. Later that year, after Qaddafi's regime was overthrown, militias from Misrata took revenge and terrorized the inhabitants of the town over their alleged loyalty to Qaddafi: Militiamen shot, raped, arbitrarily arrested and beat up civilian Tawerghas, leading to the displacement of almost the entire town. The commanders of the Misrata brigade said in 2011 that the residents of Tawergha should never return (HRW 2011).

People of Tawergha experienced multiple displacement, ranging from in-line, and cross-line to cross-border displacement: first, Tawerghas mostly fled to the Jufra region, south of Misrata, and from there to Benghazi and Tripoli (HRW 2011). Some also left Libya to Tunisia or Egypt. Since then, around 40,000 Tawerghas live displaced across Libya and have been sheltered in camps or old school buildings in and around Tripoli and Benghazi (HRW 2013). The National Transitional Council, the *de facto* government of Libya between 2011 and 2012, did not have a concrete plan for them; instead, authorities repeatedly left Tawergha camps unguarded, leading to militiamen entering the camps and committing more atrocities (HRW 2011).

Since 2012, initiatives towards national reconciliation have stalled and the displacement of the Tawerghas has remained largely unresolved. The general attitude towards them across a range of actors has been one of rejection, often paired with the idea that they should be "dumped" somewhere else. The NTC at one point suggested building "a whole city for Tawergha" near the southern oasis town of Jalo, or Sirte, Qaddafi's hometown, until national reconciliation and

return becomes possible. In the interim, high-level aid officials have advocated for a temporary solution that would improve the living conditions of the Tawergha (Aly 2011). In 2018, a reconciliation agreement between Misrata and Tawergha ended the hostility between the two cities under international auspices. According to the agreement, Tawerghas were allowed to return to their city. The Government of National Accord pledged to rebuild it and pay compensation to those affected in both cities (France24 2020). Yet, at the time of our fieldwork (2020), a large majority remained displaced and refused to return, neither believing in safe return nor in finding a rebuilt city. Interviewed Tawerghas reported the lack of protection not only in Western Libya, but also in the East (LIB17; LIB18; TUN15); interviewed directors of two Tawergha school-housed camps reported the complete absence of state support, with local charities, NGOs, IOs and INGOs filling the void and Tawerghas working as daily laborers in Benghazi to survive (LIBEX1; LIBEX12).

The trajectory of 38-year-old Farah (LIB18) serves as an enlightening example. She grew up in Tawergha in a large family with eight siblings and worked as a nurse after graduating in psychology. As a single woman, she continued living with her parents and siblings. Throughout the year 2011, the family moved houses several times trying to escape fighting in the city. In August, when the violence intensified, they fled Tawergha, first to Bani Walid, then to Tripoli where they stayed with relatives. When skirmishes approached the neighbourhood, they decided to return to Bani Walid and then left towards the south of Libya, again fleeing the approaching battles. On their way, a militia imprisoned groups of fleeing Tawerghas: "After that, about 40 cars came to us with medium weapons. My whole life I did not know weapons and never saw them, but this war taught us even to differentiate between types of weapons. Military cars came to us, armed with weapons and they told us, young men from the age of 11 years and over, go and enter the mosque and put the women in a large yard, they said to us, say, God is the greatest, God is the greatest." After 17 days, another militia entered the area and transferred a large number of Tawergha families, first to al-Jufra, a region south of Misrata, and later to Benghazi: "We came to Benghazi in October and we stayed for five days in one area, and then they told us that there are camps prepared for us, in the (anonymized) area inside Benghazi." Farah stayed in (anonymized) camp in Benghazi until 2014, when it got burned down during the war and she and her family took refuge in Ajdabiya in Northeastern Libya for seven months. They returned to (anonymized) camp after it was rebuilt and have no hope of returning to Tawergha in the near future.

On the one hand, Farah felt that with the continuing violence committed by Misrati militias, the perception of Tawerghas among the Benghazi population improved over the years: "When we first arrived, we felt that we were not wanted in Benghazi (…), they used to tell us everywhere, you did this and that but in fact we didn't do anything of what they said about us, it was racism. (…) In 2013, when the events of the Gargour area occurred in Tripoli [violent events between a Misrata militia and Tripolitanian militia], at that time, society's view of us changed and they began to tell us that we used to think that you were such-and-such, but now the truth became clear to us." On the other hand, her experience illustrates clearly practices of exclusion, extinction, deprivation and non-protection which can be linked to state-making strategies which aim to homogenize parts of the Libyan territory.

### Conclusion

In this paper, we have outlined mobility control practices during civil war which we derived from original data collected in Tunisia, Libya, Lebanon, and Syria. On this basis, we have identified three mechanisms of mobility control: First, forcing people perceived as a threat to

exit a region or state, or actively deporting them cross-line; second, disincentivizing the return of displaced people who could be a threat through practices of selective return; and third, leaving the mobility of some groups of the population mostly unobstructed, but without guaranteeing protection or services, in what we call practices of strategic laissez-faire. We also demonstrated that, for some of these practices, cooperation with state and non-state actors in neighbouring countries is crucial, in particular when it comes to practices of irregularization, taming and laissez-faire. Forcing exit and selective return across international borders necessarily depends on the aligning interests of sending and receiving states.

We have argued that these mobility control mechanisms function as attempts to (re)make the state by disposing of a part of its citizenry as a strategy of neutralising any perceived dissent in (former) opposition strongholds, or to get rid of former regime supporters in case of a regime change. Having said that, attempts to change the political community can be about changing the nature of, but also about sustaining the state. We have shown that the questions of who can move, who can return, and who cannot, are central to mobility control practices in civil war settings. This means that mobility control and how it interacts with state-making during and after civil war becomes a key question of conflict resolution and peacebuilding, as it decides who belongs to a surviving state or who will be part of a new, state-like formation, and with that, who will be part of conflict resolution and peacebuilding efforts. If during civil war, mobility control practices lead to a 'homogenisation' of a population, this means the exclusion of a formerly integral part of that same population from current and prospective 'reconstruction' efforts. What does this mean for the future relationship between the state and its citizens? Which role do old and new local elites play in such processes, and who benefits from such practices of mobility control? We hope to have made a first step to better understanding these dynamics; our typology can be a starting point for future research on civil war sending states in different world regions.

We have also given first insights under which conditions which mobility control mechanisms prevail, arguing that in rebelocracies, in which monopolies clash and identifying potential opponents and supporters is crucial for both state actors and insurgents, mobility control mechanisms of forcing exit and selective return are often the norm. We also showed that in aliocracies – spaces where rebels only intervene minimally to maintain their monopoly over the use of violence –, forcing exit and laissez-faire mechanisms are more common. Future research can further theorize and systematize the causes of variation of mobility control mechanisms in different types of civil wars with different types of actors dominating. Future research could also link our ideas to an extended analysis of the mechanisms at work in the provision of assistance and humanitarian aid. While we focused in this paper specifically on mobility control and not the provision of aid, our data clearly shows that there are selective and assortive processes at work in how different state and non-state actors manage and organise humanitarian aid to different groups of the displaced.

**Funding**

This project has received funding from the European Union's Horizon 2020 research and innovation programme under grant agreement No. 822806 (MAGYC) and under the Marie Skłodowska-Curie grant agreement No. 748344 (SYRMAGINE). Data in Libya and Syria was collected with additional funding from the German Institute for Global and Area Studies (GIGA).

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

**Annex**

Table 1: Overview of in-depth interview respondents Libyans in Libya

| Background characteristics | |
|---|---|
| **Current city of residence** | |
| Tripoli | 7 |
| Benghazi | 12 |
| **Type of mobility and displacement** (several choices possible) | |
| External returnee | 7 |
| IDP returnee | 5 |
| IDP | 7 |
| Immobile | 4 |
| **Time of mobility and displacement** (several choices possible) | |
| 2011-2013 | 7 |
| 2014-2018 | 11 |
| 2019-2020 | 4 |
| Not displaced | 2 |
| **Original region of residence in Libya** | |
| West Libya | 6 |
| East Libya | 13 |
| **Year of birth** | |
| 1955-1959 | 1 |
| 1960-1964 | 1 |
| 1965-1969 | 0 |
| 1970-1974 | 1 |
| 1975-1979 | 5 |
| 1980-1984 | 3 |
| 1985-1989 | 2 |
| 1990-1994 | 3 |
| 1995-1999 | 2 |
| 2000-2004 | 1 |
| **Sex** | |
| Male | 7 |
| Female | 12 |
| **Educational attainment** | |
| Primary school | 1 |
| Middle school (grade 7-9) | 1 |
| Secondary school (grade 10-12) | 1 |
| University or equivalent | 16 |
| **Total** | **19** |

Table 2: Overview of in-depth interview respondents Libyans in Tunisia

| Background characteristics | |
|---|---|
| **Place of residence in Tunisia (Tunis, Sfax)** | |
| Tunis | 16 |
| Sfax | 9 |
| **Year of arrival in Tunisia** | |
| 2011-2013 | 2 |
| 2014-2018 | 11 |
| 2019-2020 | 12 |
| **Original region of residence in Libya** | |
| West Libya | 14 |
| East Libya | 7 |
| South Libya | 4 |
| **Year of birth** | |
| 1960-1964 | 2 |
| 1965-1969 | 1 |
| 1970-1974 | 2 |
| 1975-1979 | 1 |
| 1980-1984 | 5 |
| 1985-1989 | 2 |
| 1990-1994 | 8 |
| 1995-1999 | 2 |
| 2000-2004 | 2 |
| **Sex** | |
| Male | 15 |
| Female | 10 |
| **Educational attainment (enrolment)** | |
| Middle school (lower secondary, grade 7-9) | 1 |
| Secondary (higher secondary, grade 10-12) | 5 |
| University or equivalent | 19 |
| **Total** | **25** |

Table 3: Overview of in-depth interview respondents Syrians in Syria

| Background characteristics | |
|---|---|
| **Current place of residence** | |
| Damascus/Rif Dimashq | 9 |
| Aleppo/Idlib | 11 |
| **Type of displacement** (several choices possible) | |
| External returnee | 8 |
| IDP returnee | 4 |
| IDP | 9 |
| Immobile | 5 |
| **Original region of residence in Syria** | |
| Damascus/Rif Dimashq | 9 |
| Aleppo/Idlib | 11 |
| **Year of birth** | |
| 1955-1959 | 0 |
| 1960-1964 | 0 |
| 1965-1969 | 1 |
| 1970-1974 | 0 |
| 1975-1979 | 2 |
| 1980-1984 | 1 |
| 1985-1989 | 2 |
| 1990-1994 | 6 |
| 1995-1999 | 7 |
| 2000-2004 | 1 |
| **Sex** | |
| Male | 8 |
| Female | 12 |
| **Educational attainment** | |
| Primary school | 0 |
| Middle school (grade 7-9) | 0 |
| Secondary school (grade 10-12) | 4 |
| University or equivalent | 16 |
| **Total** | **20** |

Table 4: Overview of in-depth interview respondents Syrians in Lebanon

| Background characteristics | |
|---|---|
| **Place of residence in Lebanon (Beirut and suburbs, Shtora)** | |
| Beirut and suburbs | 17 |
| Shtora | 7 |
| **Time of arrival in Lebanon** | |
| Before the war | 1 |
| 2011-2013 | 11 |
| 2014-2018 | 12 |
| 2019-2020 | 0 |
| **Last place of residence in Syria before emigration** | |
| Damascus | 4 |
| Rif Dimashq | 10 |
| Hama | 2 |
| Homs | 3 |
| Rif Aleppo | 2 |
| Aleppo | 2 |
| Daraa | 1 |
| **Type of displacement** (several choices possible) | |
| Externally displaced | 24 |
| IDP | 15 |
| External returnee | 1 |
| Migrant | 1 |
| **Year of birth** | |
| 1950-1954 | 1 |
| 1960-1964 | 0 |
| 1965-1969 | 0 |
| 1970-1974 | 1 |
| 1975-1979 | 1 |
| 1980-1984 | 4 |
| 1985-1989 | 7 |
| 1990-1994 | 9 |
| 1995-1999 | 1 |
| 2000-2004 | 0 |
| **Sex** | |
| Male | 9 |
| Female | 15 |
| **Educational attainment (enrolment)** | |
| Primary school | 5 |
| Middle school (grade 7-9) | 3 |
| Secondary school (grade 10-12) | 2 |
| University | 14 |
| **Total** | **24** |

**Table 5: Overview expert interviews**

| Libya | |
|---|---|
| Tarhouna IDP camp | Libya |
| IOM Libya | Tunisia/virtual |
| Libyan Red Crescent | Libya/virtual |
| Libyan Ministry of Displacement and IDP Affairs | Libya/virtual |
| ICRC Libya | Libya/virtual |
| UNHCR Libya | Libya/virtual |
| Libyan Red Crescent | Libya/virtual |
| UNHCR Libya | Tunisia/virtual |
| International Medical Corps Libya | Libya/virtual |
| Danish Refugee Council Libya | Tunisia/virtual |
| OCHA Libya | Libya/virtual |
| Tawergha IDP camp | Libya |
| IOM Libya / DTM | Libya/virtual |
| INGO Forum Libya | Tunisia/France/virtual |
| UNHCR Tunisia | Tunisia/ virtual |
| Terre d'Asile Tunisia | Tunisia/ virtual |
| **Syria** | |
| Syrian Arab Red Crescent | Syria/virtual |
| Norwegian Refugee Council | Syria/virtual |
| Namaa Developmental Association | Syria/virtual |
| Durable Solutions Platform | Jordan/virtual |
| Jesuit Refugee Service | Syria/virtual |
| North-East-Syria NGO Forum | Syria/virtual |
| Syria INGO Regional Forum | Jordan/virtual |
| Norwegian Refugee Council/DSP | Lebanon/virtual |
| Baytna | Turkey/virtual |
| Syrian Center for Policy Research | Europe/virtual |
| Syrian Association for Citizens' Dignity | Turkey/virtual |
| UNHCR | Jordan/virtual |
| EU Mission to Syria | Lebanon/virtual |
| Lebanese Ministry of Social Affairs | Lebanon/virtual |
| International Humanitarian Relief | Syria/virtual |