# Peer review of "Mobility Control as State-Making in Civil War: Forcing Exit, Selective Return and Strategic Laissez-Faire"

_Migration Politics_

## Round 1 · Referee Report · Anonymous (Referee 1) · 2022-6-11

Strengths

The paper has key strengths in:

1) Novel Data 2) Coverage of literatures 3) Attempting to bring together two aligned but rather different fields and research traditions 4) Useful cases and elements of comparison 5) Highlighting opportunities for future research

Weaknesses

The paper has weaknesses in:

1) Lack of depth and full clarity on methodology 2) Fully marketing its ideas 3) Some minor gaps in literature which offer opportunities for more depth 4) Some structural/signposting issues - the empirical materials in the section before the conclusion are not deployed or extrapolated to best effect

Report

This article has a lot of potential and with some polish can absolutely be published. Most of the changes are either for clarity or are intended to make better use of the data. One of the key changes however is to strengthen the methodology section which is at present too limited and descriptive. Another key change is to make better use of the micro level cases from the data - linking them more explicitly to the previous section and into the conclusion itself (the latter sells the article short in some ways and could do with ending on a different note). I would also suggest a few other sources which might be profitably consulted. In essence this is a minor R&R which just requires a bit of polish the data are good and the article is well put together so very close to being publishable already.

Requested changes

I have a fully annotated document (handwritten) which includes both stylistic and substantive changes which the authors might wish to take into consideration.

---

## Round 1 · Referee Report · Anonymous (Referee 2) · 2022-7-9

Report

General comment
The paper is interesting and definitely worth publishing, but a little bit disappointing as the empirical material (that is flagged out in the introduction) is not mobilised as it could.
More importantly, I am doubtful of the general argument: mobility control as state making. I think there is value in understanding mobility as a “policy” (to be defined!!) in civil war, used by different groups, but I am not convinced by the “state making “argument – I think the notion of “State” as it is commonly understood is rather misleading in the context of Syria, and possibly of Libya, and in particular with regard to forced mobility/immobility.
If I were the authors, I would either develop this argument robustly or rather focus the paper on mobility control and suggest a typology and use the empirical material to illustrate this typology. That would be an analytical proposition interesting enough without needing this unconvincing argument of state making, that is too weakly developed in the paper, as far as for my reading.
The value of the comparison is not striking in the paper. It feels more a juxtaposition that two examples that help to understand a question… This could be better intertwined. Syria is the main example and the one that is the most convincing with regard to the research question…

Here are detailed remarks on different sections.
Introduction
A clearer stance could be made on definitions: the terms of migration/ mobility / movement are used as synonyms, whereas they are not. A better definition of the concepts involved in the paper could improve the analytical accuracy of the paper.
Also, clarification on the central argument of the paper could be made clearer:
Why controlling mobility is interesting for actors involved in a civil war? More importantly, why “demographic engineering” is an instrument of state making (this is stated rather than argued)? Is mobility control the same thing as population control (on one’s territory)? What is more important for an actor in a civil war: controlling mobility or controlling the population? Does mobility control really happen?
Definition of the object: one can see what mobility control involves when an area is besieged, or a border fenced and militarized, but otherwise?
The paper gives the impression that actors are actually able to control – is its really the case in general (even in non civil war contexts, see the issue of illegal migration) and in Syria and Libya in particular? There is something to be critically discussed more thoroughly than to cite Betts and his “implicit forms of governance” here.
I would also mention the fact that talking about State with regard to the Syrian regime is a bit of a fiction: it is more a mafia supported by militias and businessmen, and most importantly by Russian and Iran, rather than a State – a note on this may be needed, as otherwise one has a false representation of what the actors at play are – a similar situation of State degradation in Libya.
1. With regards to Syria, its borders (porous and with different border regimes), mass displacement as a weapon of war, and linking internal/external displacement, the authors may want to look at Vignal, 2017, “The Changing Borders and Borderlands of Syria in a Time of Conflict”, International Affairs, 93/4, 809—827. Chapters 2, 5, 6, 8 and 9 of Vignal, 2021, War-Torn, The unmaking of Syria (2011-2021) are relevant on mobility control.

First section
The section properly explains why mobility control is important for actors in a civil war and defines all the terms involved BUT fails to define mobility on the one hand, and mobility control on the other – or at least saying there that the paper precisely aims at exploring which type of mobility control takes ace in civil war settings, and why (to reinforce the actors operating mobility control).

Section 2
Also, it would be better to avoid terms such as “boundaries” that have a double meaning: territorial one and abstract one. Page 6 for instance: the “boundaries of identity”: ambiguous – territorial or political? Important to distinguish the territorial and the abstract meanings when you discuss actual mobility, ie with a territorial implication. You could use here “set the limits” or “definition” etc.., of belonging…”.
Page 7: Considering the ambition of the paper, it is surprising that the focus is (in addition to the displaced people) on “the community of practitioners consists of policymakers and international stakeholders involved in migration governance and humanitarian aid”: I would have expected that the authors were looking at the “local” actors? Needs clarification.
I am not sure that the definition given of practices (pouliot) is crystal clear here.
What is “practices tracing”? this is not explained – readers need to be explained and explained as well why this method is relevant for your analytical objective.

Section 3:
Table 1 should be put just after it is mentioned (top of page 8) otherwise this quite abstract paper becomes difficult to follow.
Also, related to the definition of practices, when looking at the table, one understands “practices” in their more general meaning – this needs to be clarified as it confuses the reader whereas it should not -- it is quite straightforward.
Authors may be interested in a discussion of terms displaced refugees, nazihin/muhajerin etc), in Lebanon by Aubin-Boltanski and Vignal ( “Hosting and Being Hosted in Times of Crisis: Exploring the Multi-layered Patterns of Syrian Refuge in the Dayr al-Ahmar region, Northern Bekaa, Lebanon”, in Mobility and Forced Displacement in the Middle East).
With table 1, one understands better what is understood as mobility control: a few examples and the mention of table 1 could be introduced in the introduction.
Page 10: the border with Lebanon is not closed as far as I am aware? And also: one would need to distinguished between securitized border (walled etc, such as Tur/syr) and very porous one (LEB/SYR). Also, distinguish between legal and illegal crossing…
The paper does not explain why and how the authors chose this typology (these three mechanisms).

Page 11:
A sentence on how many people do return would be a useful contextualisation (very few).
“All governing actors have a return policy: goS, SDF, HTS, but we see no return, so (…) they are completely failing at implementing their return policy, which makes you question – do they really have a return policy, or have they just written a policy to appeal to their patron, or to their sponsor? So, does the Syrian government really want Syrians to return, or are they just doing it because Russia told them to do it? And does the Syrian opposition really want Syrians to return to its areas or are they doing it just because Turkey told them to do it?”: the authors validate their analysis with the use of a quotation, that is the analysis of a Syrian respondent rather than a scientific analysis – more significant if this quote was presented as the way in which some sections of the society see the implementation of return policies ….

P12: practices of taming do not need strong institutions in my view…. Ie it is not the strength of institutions that makes them successful. Here, I link with my previous remarks in the notion of State versus non state , and the fact that the Syrian state is pretty dismantled. And the Hezbollah, on which the paragraph expands, is not a State group…P16 (but this is often the case elsewhere) we need dates, there as in other places, otherwise the conflicts feel like ahistorical streams…… (“After Ramadan, we returned to the area of Jabal al-Zawiya, the…” then At the same time, the region has become a refuge for IDPs fr …“ )
“demographic engineering” is used by Syrians in many different ways and with regard to many different actors. As a not a scientific notion -- I think it would be better here to explain that those policies are perceived as demographic engineering – the authors develop their own analytical tools, why do they not use them here?

---

## Round 2 · Author Response

Dear Dr. Thiollet, dear reviewers,

We would like to thank you for offering us the chance to improve our manuscript based on the extensive and constructive feedback of the reviewers and your structured suggestions, which helped us to bring this paper forward.

As you summarized in your email, Hélène, our revisions should focus on making a more convincing argument in favour of our theoretical claim (civil war/mobility/state making) and on better delineating its limitations, on strengthening the comparative design, and on drawing upon the critical literature and empirical information on the sociology of the “state” in contemporary Libya and Syria. We slightly cut the paper in length and better streamlined our argument across the paper. We decided to restructure the paper and incorporate parts of section 4 in section 3 to illustrate how the different mobility control mechanisms play out and overlap in specific localities in Syria and Libya.

In the below letter, we respond in detail to each point raised by the two reviewers and explain how we accommodated the feedback in question. We completely revised the document, which is why we chose not to submit the document in track change mode for better readability. We copy our modifications in italics in the table below if they were not too long, otherwise we refer to the page numbers of the revised document.

We hope that the revisions sufficiently address your suggestions and are looking forward to your answer.

Best wishes
Christiane (Fröhlich) and Lea (Müller-Funk)

---

## Round 2 · List of Changes

Reviewer 1
We are very thankful for the reviewer’s thoughtful and rich reflections. This critical and detailed review has helped us to strengthen our core contribution, our research design and our analysis and to subsequently develop a core argument across the paper. We are also grateful for the many stylistic and language suggestions, which we incorporated directly in the text.

Comment: 1. Lack of depth and full clarity on methodology. How do you get to the mechanism grouping?
Many thanks for this valid criticism to better clarify our methodology. We have elaborated on practice tracing and the abstraction process leading to our mechanisms, as well as on our sample, in the introduction and the methodology section. We also included a paragraph about the limitations of our methodology. The modifications are too extensive to copy here.

Comment: 2. Fully marketing its ideas
Thank you for this comment. We completely rewrote the introduction and the conclusion to bring our contributions to the literature to the fore. The modifications are too extensive to copy here.

Comment: 3. Some minor gaps in literature which offer opportunities for more depth
Many thanks for this valid criticism. We have consulted the recommended literature and included most of it in the current version. We had to cut some other references to make space for these inclusions. The modifications are too extensive to copy here.

Comment: 4. Some structural/ signposting issues - the empirical materials in the section before the conclusion are not deployed or
extrapolated to best effect.
Many thanks for this valid criticism. We decided to cut section 4 and include the vignettes within the mechanisms to illustrate how the different mobility control mechanisms play out in a specific locality and the ways in which people have reacted and resisted to such attempts to control mobility. The modifications are too extensive to copy here.

Comment: 5. Quite a lot to take in in the first paragraph Thanks for this comment. We tried to improve readability of the first paragraph and many other paragraphs in the paper by completely rewriting the introduction. The modifications are too extensive to copy here.

Comment: 6. How is state-making different from state formation? Does it differ between types of “stateness”/actors?
Thanks for this comment. We use state-making because some of the actors we are looking at are fully formed states and thus state formation does not seem to be the right term for their efforts to remain or become a fully formed state again. Other actors, e.g. HTS, could indeed be described as engaging in state formation. We chose the term state-making to encompass both types of “stateness”, and clarified this in the text (Footnote 2):
"We use the term ‚state-making‘, rather than ‘state formation’, to signify both attempts by fully fledged states to remain a functional state or regain full functionality, and attempts by non-state actors to become more state-like."

Comment: 7. Why are Syria and Libya so emblematic?
Thanks for this important comment. We now clarified the value of our two cases in the introduction (p. 1) and in the methodology section on p. 6: "It draws on qualitative empirical material from two internationalized intrastate wars which have triggered mass displacement within and across borders—the civil wars in Syria and Libya since 2011. Syria and Libya are emblematic cases of contemporary civil war states given the politically organized, large-scale, sustained violent conflicts that have occurred within their territories over the past ten years. They are characterised by an uneven state presence, with the Syrian regime continuing to govern after having re-conquered large, but not all parts of the country at the time of writing, and Libya consisting of two separate parts with competing governments after the overthrow of the Qaddafi regime.
In Syria, the regime survived but has to share power with a range of different actors (Vignal 2017, 814), while in Libya, two competing governments have been in place. While both are contemporary cases of internationalised civil wars with large numbers of internally and externally displaced people, possibilities to exit the country and flee across borders diverge considerably. Since 2014, it has become increasingly difficult for Syrians to enter neighbouring Jordan, Lebanon, and Turkey. Lebanon, for example, changed from a mutual mobility agreement which was still respected at the beginning of the Syrian conflict to one characterised by border closures and political actors advocating for a return of Syrian refugees. In contrast, the Libyan-Tunisian and Libyan-Egyptian borders have mostly remained open for Libyans. Tunisia, for example, adopted a laissez-faire approach towards Libyans tolerating their entry and presence without providing refugee status."

Comment: 8. For what purpose do you interpret mobility control as strategy of state-making?
Thanks for this important comment. By restructuring the text, we now explain our reasoning already in the introduction for better readability and understanding (p. 3): "Civil wars present opportunities to reconfigure social contracts and power constellations. War often weakens, and sometimes destroys, state institutions; it can also facilitate the emergence of local orders with actors like religious authorities, tribes, rebels, and clans attempting to exercise state functions (Khalaf 2015; Arjona 2014, Worrall 2017, Migdal 2001, Staniland 2012). This often results in hybrid, inconsistent, and constantly fluctuating regimes of control. In Syria’s civil war, for example, there is rarely a unique and cohesive authority that oversees different functions but rather a diversity of control and overlapping regimes (Vignal 2017). At the same time, civil wars, especially internationalized ones, produce the largest refugee populations (Schmeidl 1997), leading to mobility control becoming a central field in which actors attempt to exercise state functions.
In this paper, we call attempts to exercise state functions strategies of state-making. These include attempts to fulfil government-like functions such as protection, justice, passing laws, raising taxes, the provision of basic services, and mobility control. In this paper, we specifically focus on attempts to monopolize the right to control and regulate movement. With this, we follow migration scholars who have long argued that controlling the movement of people across both internal and external borders is crucial for state formation and consolidation (Torpey 1997; Torpey 2000; Zolberg 1978; Zolberg 2008; Vigneswaran and Quirk 2015, McKeown 2008)."

Comment: 9. Does state-making include efforts to establish law and order? Thanks for this important comment. We included this aspect on p. 4, see: "Once control of a geographical location is established, other efforts to create a state might follow, such as building up institutions, providing social services (McColl 1969; Stewart 2018), establishing ‘law and order’, or humanitarian assistance to the displaced. In fact, who provides for the displaced is often a key question of national sovereignty in conflicts (Rahal and White 2022) and generates ‘rents’."

Comment: 10. Libya seems to be consistently quite marginal.
Thanks for this. We have clarified in the methodology section that the paper does not aim for a strictly comparative design, we employed Syria as our main case and Libya as our complementary case, see p. 6: "Our comparative approach is explorative, insofar as we mostly draw on the Syrian case and complement and contrast our analysis through cross-case insights from Libya. We use the concept of mechanisms for the theoretical abstractions we coin to classify these practices across cases. As Pouliot (2014, 238) writes, “mechanisms are analytical constructs whose objective is not to match actual social instances, but to draw useful connections between them”. "

Comment: 11. Explain why state actors have employed strategic displacement in two thirds of civil wars between 1945 and 2008, but not in the other third
Thank you for this important point. We argue that the result would be different if strategic laissez-faire had been considered. This finding is now part of our analysis and the conclusion, see p. 17: "We identified three mechanisms of mobility control: Forcing people to exit a region or state, or actively deporting them cross-line; disincentivizing the return of displaced people through practices of selective return which ultimately constitutes a form of expulsion for some groups; and leaving the mobility of some groups mostly unobstructed, but without guaranteeing protection or services, in what we call practices of strategic laissez-faire. In fact, including the absence of regulation in an analysis of mobility control can maybe explain why previous research may have underestimated the extent of strategic displacement in civil war countries (e.g. Lichtenheld 2020)."

Comment: 12. Expand and rephrase the conclusion, sell paper better
Thank you for this comment. We have expanded the conclusion considerably and outlined avenues for future research as well as addressed the limitations of our own work. The modifications are too extensive to copy here.

Comment: 13. Analyse more why Libyan officials have referred to externally displaced Libyans as ‘migrants’
Thank you for this comment. We have added a few sentences to explain why this practices is depoliticizing displacement and downplaying protection needs and included a quote from a Libyan displaced person to contrast these perceptions, see p. 9: "Libyan officials and Libyan reports referred to externally displaced Libyans as “Libyan migrants” (LIBEX4), thereby depoliticising displacement and downplaying protection needs. After all, Libyan “refugees” would have access to a different set of rights and assistance than “migrants”, who supposedly left of their own accord. A Libyan woman living in Tunis summarized the approach of the Libyan state(s) as follows, “Our government isn’t offering us the needed services and we face problems that we have no one to trust that can represent us or protect us because the embassy itself isn’t cooperating […]. Our government or representatives supposedly or even the people who speak for us are not helping at all as if we have no one […]. So we the people who live here have to take care of ourselves and with the few social connections that we made” (TUN20)."

Comment 14: Show more clearly how different practices can interlink to produce a certain outcome (p.11)
Thanks for this valuable comment. We have included the following sentences on p. 12-13: "Here, practices depriving Syrians of access to their private property are perpetuating practices preventing legal return, together forming a continuum of deprivation and irregularisation. Abboud (2020) has similarly argued in this regard that Syrian citizenship is increasingly bifurcated along the lines of settled or reconciled and rejected citizens."

Reviewer 2
We are very thankful for the reviewer’s conceptual reflections and critical comments about the empirical analysis, especially about our data from Syria. The suggestions for revisions helped us to better carve out our contribution.

Comment: 1. empirical material is not mobilised as it could and the agency of displaced people should be underlined more
Many thanks for this valuable comment. Following your suggestion, we have included empirical examples throughout the text to better explain our research question and the research gap we are addressing. We have also highlighted in the introduction and theory section that mobility control is not necessary successful and emphasized that people have agency in reacting to / resisting practices of mobility control. We also restructured the vignettes, which were not used to their full potential, to better illustrate how people react to mobility control practices to underline their agency. We included more quotes of our respondents who have been confronted with these mechanisms. The modifications are too extensive to copy here.

Comment: 2. The paper does not explain why and how the authors chose this typology (these three mechanisms)
Many thanks for this valid criticism to better clarify our methodology. We have elaborated on practice tracing and the abstraction process leading to our mechanisms, as well as on our sample, in the introduction and methodology section. We also included a paragraph about the limitations of our methodology. The modifications are too extensive to copy here.

Comment: 3. doubtful of the general argument: mobility control as state making. I think there is value in understanding mobility as a “policy” (to be defined!!) in civil war, used by different groups, but I am not convinced by the “state making “argument – I think the notion of “State” as it is commonly understood is rather misleading in the context of Syria, and possibly of Libya, and in particular with regard to forced mobility/immobility. I would either develop this argument robustly or rather focus the paper on mobility control and suggest a typology and use the empirical material to illustrate this typology. That would be an analytical proposition interesting enough without needing this unconvincing argument of state making, that is too weakly developed in the paper, as far as for my reading.
Many thanks for this valuable suggestion. We take your critique of us seeing state-making as too monolithic very seriously. We have included more critical and empirical literature to illustrate the hybrid and complex nature of state-making actors in our cases throughout the text, but esp. on p. 5. We have also included a definition of mobility and mobility control on p. 5. The revisions are too extensive to copy here.

Comment: 4 The value of the comparison is not striking in the paper. It feels more a juxtaposition that two examples that help to understand a question. This could be better intertwined. Syria is the main example and the one that is the most convincing with regard to the research question.
Many thanks for this suggestion. We made it clear that Syria is our main case and Libya plays a complementary role in our research design, on p. 6: "Our comparative approach is explorative, insofar as we mostly draw on the Syrian case and complement and contrast our analysis through cross-case insights from Libya. We use the concept of mechanisms for the theoretical abstractions we coin to classify these practices across cases. As Pouliot (2014, 238) writes, “mechanisms are analytical constructs whose objective is not to match actual social instances, but to draw useful connections between them”."

Comment: 5 Do not use “boundaries”
Thank you for this comment. We have substituted the term, which we used twice in the text, with the alternative formulations as suggested.

Comment: 6 Definition of the object: one can see what mobility control involves when an area is besieged, or a border fenced and militarized, but otherwise?
Thank you for this comment. We have added a paragraph about the limits of visibility of mobility control, see p. 6: "We use practice tracing as many practices related to mobility control in Syria and Libya are not based on formal policy or public agreements, but are invisible, opaque, informal, and locally (re)negotiated. The basic objective of practice tracing is to understand what a practice counts as in the situation at hand and to move beyond singular causality toward cross-case insights. Our comparative approach is explorative, insofar as we mostly draw on the Syrian case and complement and contrast our analysis through cross-case insights from Libya. We use the concept of mechanisms for the theoretical abstractions we coin to classify these practices across cases. As Pouliot (2014, 238) writes, “mechanisms are analytical constructs whose objective is not to match actual social instances, but to draw useful connections between them”. Importantly, even when practices cannot be seen, they may be talked about through interviews or read thanks to textual analysis (Pouliot 2014, 246)."

Comment: 7. I would also mention the fact that talking about State with regard to the Syrian regime is a bit of a fiction: it is more a mafia supported by militias and businessmen, and most importantly by Russian and Iran, rather than a State – a note on this may be needed, as otherwise one has a false representation of what the actors at play are – a similar situation of State degradation in Libya.
Thank you for this important comment. We have included this aspect in the revised introduction, methodology and analysis section. The revisions are too extensive to copy here.

Comment: 8. Section 1. define mobility on the one hand, and mobility control on the other – or at least say that the paper precisely aims at exploring which type of mobility control takes place in civil war settings, and why (to reinforce the actors operating mobility control).
Thank you for this valuable comment. We have included a section in the revised text explaining that we aim to understand which types of mobility control take place in civil war settings, and why. We also defined mobility and mobility control, see p. 3-4: "With our analysis, we aim to understand which types of mobility control take place in civil war settings, and which role they play in state-making efforts of different actors. The mobilities literature focuses on the ‘politics of mobility’, i.e., the socio-spatial inequalities that are (re)produced by differential access to or effects of various kinds of mobility (Cook and Butz 2018). While mobility is a fundamental aspect of daily life for people everywhere, access to mobility is often experienced unequally along lines of gender, ethnicity, race, religion, age, and social class. Control over mobility is therefore a form of power with deep historical roots (Sheller 2018, 24-25).
We define mobility control, building on Zolberg’s (1978, 243) and Natter’s (2019, 31) work, as (i) practices around formal policies, laws, and regulations governing internal and external border control, entry, and exit regulations; (ii) informal dynamics (for example, differences between administrations and localities); and (iii) laissez-faire, with the purposive absence of regulation. The rationale for including the absence of regulation is motivated by our wish to understand implementation gaps, degrees of lawlessness, and legal vulnerability along the full continuum of movement in civil war settings. We also see the societal and political negotiation of displacement terminology—in other words, processes of labelling mobile populations—as part of mobility control, because different terms indicate different reactions to displacement (Zetter 2007; Erdal & Oeppen 2017; Aubin-Boltanski & Vignal 2020).
Mobility control in civil war is characterised by a multiplicity of actors, and shifts according to changing power dynamics."

Comment: 9. Add dates to clarify the timeline
Thank you for this comment, we have added dates wherever possible.

Comment: 10. Considering the ambition of the paper, it is surprising that the focus is (in addition to the displaced people) on “the community of practitioners consists of policymakers and international stakeholders involved in migration governance and humanitarian aid”: I would have expected that the authors were looking at the “local” actors? Needs clarification.
Thank you for this comment. We have explained our sample choice more extensively in the methodology section of the revised text, p. 6-7, see: "Methodologically speaking, the fact that practices describe ways of doing things that are known to practitioners means that practices must be understood from within the community of practitioners to restore the intersubjective meanings that are bound up in them. In our case, the community of practitioners consists of policymakers and stakeholders involved in migration governance and humanitarian aid on the local and national level. These policymakers and stakeholders can be Syrian or Libyan nationals (e.g., mayors, members of the Autonomous Administration of North and East Syria (AANES) etc.), or internationals based in the countries (e.g., representatives of international organisations). Our study also includes the perspective of the governed, i.e., displaced people, who have experienced different types of mobility control."
We now also discussed the limitations of our data better on p. 7, see: "Our data and analysis have clear limitations. Given the volatile character of civil war, our analysis only provides geographical and temporal snapshots. We do not claim to comprehensively study all mobility control practices taking place in civil wars, nor all state and non-state actors present in both countries. Our data reflects our limited access to different localities due to security concerns and Covid-19 regulations. Some regions remained inaccessible to us and some virtual interviews with participants living in Libya and Syria avoided certain sensitive political topics to safeguard our respondents. Our expert interviews have a bias towards stakeholders in international organisations and international NGOs, as it was extremely difficult to engage with national and local political actors virtually during the pandemic. Our typology thus should be understood as a starting point for reflection to which future research can add."

Comment 11: With table 1, one understands better what is understood as mobility control: a few examples and the mention of table 1 could be introduced in the introduction.
Thank you for this valuable comment. Following your recommendation, we have moved the table to page 7 of the paper where we explain our analytical strategy.

Comment: 12. With regards to Syria, its borders (porous and with different border regimes), mass displacement as a weapon of war, and linking internal/external displacement, the authors may want to look at Vignal, 2017, “The Changing Borders and Borderlands of Syria in a Time of Conflict”, International Affairs, 93/4, 809—827. Chapters 2, 5, 6, 8 and 9 of Vignal, 2021, War-Torn, The unmaking of Syria (2011-2021) are relevant on mobility control.
Thanks for this – we have included the literature recommendations in the revised text.

Comment: 13. A sentence on how many people do return would be a useful contextualisation (very few).
Thank you for this comment, we have added information about the marginal magnitude of return movements in the Syrian case, see p. 11: "It is important to note that the number of returnees remains low despite the Syrian regime and its supporters (Russia, parts of Lebanon) peddling a discourse of return."

Comment: 14. Referring to a quote by SYREX9 on p. 11: the authors validate their analysis with the use of a quotation, that is the analysis of a Syrian respondent rather than a scientific analysis – more significant if this quote was presented as the way in which some sections of the society see the implementation of return policies.
Thank you for this valuable comment. We have rephrased the section to present the quote in the way you suggested.

Comment: 15. Why controlling mobility is interesting for actors involved in a civil war? More importantly, why “demographic engineering” is an instrument of state making (this is stated rather than argued)?
Thank you for this comment. We have used the vignette about Jabal al-Zawiya to explain why controlling mobility is interesting for civil war actors, and have moved it up to the section on the mechanism of selective return. We have also made clear that “demographic engineering” is a term used by some of our respondents to describe the practices and mechanisms we have observed. (p. 12f). The modifications are too extensive to copy here.

You are currently on this page

Resubmission scipost_202204_00014v2 on 20 October 2022

---

## Editorial Decision

unknown